# Fungal *X*-Intrinsic Protein Aquaporin from *Trichoderma atroviride*: Structural and Functional Considerations

**DOI:** 10.3390/biom11020338

**Published:** 2021-02-23

**Authors:** Maroua Ben Amira, Mohamed Faize, Magnus Karlsson, Mukesh Dubey, Magdalena Frąc, Jacek Panek, Boris Fumanal, Aurélie Gousset-Dupont, Jean-Louis Julien, Hatem Chaar, Daniel Auguin, Robin Mom, Philippe Label, Jean-Stéphane Venisse

**Affiliations:** 1Université Clermont Auvergne, INRAE, PIAF, 63000 Clermont-Ferrand, France; marouabenamira@gmail.com (M.B.A.); boris.fumanal@uca.fr (B.F.); aurelie.gousset@uca.fr (A.G.-D.); j-louis.julien@uca.fr (J.-L.J.); Robin.mom@uca.fr (R.M.); philippe.label@inrae.fr (P.L.); 2Faculté des Sciences de Bizerte, Zarzouna 7021, Tunisia; 3Laboratory of Plant Biotechnology, Ecology and Ecosystem Valorization, Faculty of Sciences, University Chouaib Doukkali, El Jadida 24000, Morocco; faizemohamed@yahoo.fr; 4Uppsala BioCenter, Department of Forest Mycology and Plant Pathology, Swedish University of Agricultural Sciences, SE-75007 Uppsala, Sweden; Magnus.Karlsson@slu.se (M.K.); Mukesh.Dubey@slu.se (M.D.); 5Institute of Agrophysics, Polish Academy of Sciences, Doświadczalna 4, 20-290 Lublin, Poland; m.frac@ipan.lublin.pl (M.F.); j.panek@ipan.lublin.pl (J.P.); 6Crop Improvement Laboratory, National Institute of Agronomy of Tunisia (INRAT), Ariana 2049, Tunisia; chaarh@yahoo.com; 7Laboratoire de Biologie des Ligneux et des Grandes Cultures, Université d’Orléans, UPRES EA 1207, INRA-USC1328 Orléans, France; auguin@univ-orleans.fr

**Keywords:** aquaporin, uncharacterized X-Intrinsic proteins, *Trichoderma atroviride*, 3D modeling, chlamydospores, pentose phosphate pathway, stress responses

## Abstract

The major intrinsic protein (MIP) superfamily is a key part of the fungal transmembrane transport network. It facilitates the transport of water and low molecular weight solutes across biomembranes. The fungal uncharacterized X-Intrinsic Protein (XIP) subfamily includes the full protein diversity of MIP. Their biological functions still remain fully hypothetical. The aim of this study is still to deepen the diversity and the structure of the XIP subfamily in light of the MIP counterparts—the aquaporins (AQPs) and aquaglyceroporins (AQGPs)—and to describe for the first time their function in the development, biomass accumulation, and mycoparasitic aptitudes of the fungal bioagent *Trichoderma atroviride*. The fungus-XIP clade, with one member (*Triat*XIP), is one of the three clades of MIPs that make up the diversity of *T. atroviride* MIPs, along with the AQPs (three members) and the AQGPs (three members). *Triat*XIP resembles those of strict aquaporins, predicting water diffusion and possibly other small polar solutes due to particularly wider ar/R constriction with a Lysine substitution at the LE2 position. The XIP loss of function in ∆*Triat*XIP mutants slightly delays biomass accumulation but does not impact mycoparasitic activities. ∆*Triat*MIP forms colonies similar to wild type; however, the hyphae are slightly thinner and colonies produce rare chlamydospores in PDA and specific media, most of which are relatively small and exhibit abnormal morphologies. To better understand the molecular causes of these deviant phenotypes, a wide-metabolic survey of the ∆*Triat*XIPs demonstrates that the delayed growth kinetic, correlated to a decrease in respiration rate, is caused by perturbations in the pentose phosphate pathway. Furthermore, the null expression of the *XIP* gene strongly impacts the expression of four expressed *MIP*-encoding genes of *T. atroviride*, a plausible compensating effect which safeguards the physiological integrity and life cycle of the fungus. This paper offers an overview of the fungal XIP family in the biocontrol agent *T. atroviride* which will be useful for further functional analysis of this particular MIP subfamily in vegetative growth and the environmental stress response in fungi. Ultimately, these findings have implications for the ecophysiology of *Trichoderma* spp. in natural, agronomic, and industrial systems.

## 1. Introduction

The *Trichoderma* species are filamentous ascomycete fungi belonging to the order Hypocreales in the class Sordariomycetes. The *Trichoderma* genus is predominantly composed of green-spored and filamentous soil-dwelling fungi found worldwide, which is, above all, cosmopolitan and ubiquitous in the environment. Like any chemo-heterotrophic organisms, they obtain their reduced carbon source from other organisms whether they are living or dead. With the exception of the pathogenic and biotrophic behavior of a few specific *Trichoderma* strains [1,2,3,4], *Trichoderma* strains are mainly saprophytes and necrotrophic mycoparasites. Even more promising, and notably with plants, a few strains are part of symbiotic and/or commensal relationships where they establish a robust and long-lasting colonization of organ surfaces with possible asymptomatic penetrations into the epidermis and a few cells below this level [5,6]. As a result, by acting as plant symbionts (endophytism), specific *Trichoderma* strains have been extensively developed as biocontrol agents against different plant diseases due to their ability to successfully antagonize plant pathogenic fungi or bacteria [7,8,9]. Several indirect and direct coordinated strategies play substantial roles in their ability to compete with other microorganisms [9,10]. In addition, root colonization by *Trichoderma* spp. may mitigate the detrimental effects of abiotic stresses such as salinity and drought by enhancing global plant resistance, uptake and use of nutrients, root growth and development, as well as crop productivity [11,12].

The strategies of *Trichoderma* spp. deployed against antagonized phytopathogens are complex. Our understanding of the mechanisms of biocontrol and the *Trichoderma*–plant–pathogen interaction is incomplete, especially concerning nutrition. *Trichoderma* spp. display a remarkable nutritional versatility. They succeed in various heterotrophic and biotrophic interactions with a very wide range of hosts, being able to exhibit various adapted polyphagical lifestyles by decomposing dead organic matter [13]. They feed by extracellular digestion, secreting into the extracellular environment a cocktail of digestive enzymes that break down biopolymers (cellulose, hemi-cellulose, chitin, lignin, lipids, proteins, etc.) [14]. The resulting external soluble products are subsequently absorbed into the fungal cells by transmembrane diffusions common to all fungi. This route requires an arsenal of integral membrane proteins, channels and transporters, such as a large array of sugar transporters (major facilitator superfamily (MFS)) [15], nitrogen transporters (di/tri-peptide transporters (PTR)) [16], and ATP-binding cassette (ABC) transporters [17]. However, most potential systems for membrane transport proteins encoded by filamentous fungi have not yet been characterized, including the major intrinsic protein (MIP) superfamily, also called aquaporins (AQPs) [18]. These channels facilitate the transport of water and a variety of small molecules across biological membranes, the plasmalemic and the endomembrane system, of unicellular and multicellular organisms [19]. In filamentous fungi, preliminary evidence shows that MIPs are involved in the regulation of hyphal growth, sclerotia formation, conidiation, spore germination, virulence, secondary metabolisms, and the transport of various molecules [20,21,22,23,24,25]. In this respect, the impact of transcriptional and protein regulation of MIPs on the fungus’ life cycle remains a fascinating area that deserves further exploration.

The MIP superfamily is present in all kingdoms from Prokaryotes (archaea and eubacteria) to Eukaryotes (fungi, insects, animals, and plants) [19]. Most of these clades share two subfamilies: the aquaporins (AQPs, *lato sensu*), and the aquaglyceroporins (AGPs or AQGPs), which distinguish themselves by their primary sequences and their ability to transport various solutes. The aquaporins *lato sensu* encompass both the “orthodox” or “classical” aquaporins (AQPs, *stricto sensu*), which are mainly water channels. The aquaglyceroporins additionally transport small organic compounds such as urea and glycerol due to differential hydrophobic and pore size features. Fungi possess both of these two MIP prototype groups in which four distinct sub-groups can be distinguished based on their primary sequences: the orthodox fungal water channels, the facultative fungal aquaporins, the fungal aquaglyceroporins, and the fungal uncategorized (*X*)-intrinsic proteins (XIPs) [26]. This fourth group, XIP, resembles orthodox aquaporins. It is predominantly shared between certain clades of fungi and Viridiplantae, and marginally present in some protozoa [26,27,28,29,30,31]. So far, XIP has been found to be the only MIP subgroup common to both plants and mycetes, implying a particular evolutionary relationship between these two unrelated phylogenetic lineages that needs to be resolved. In addition, this suggests specific roles in solute transport and/or protein regulation of these channels which deserve better understanding.

Like any other MIP, the XIP protein protomer structure is highly conserved and resembles an “hourglass model” [29,30,32]. In plants, several *XIP* copies per genome constitute the XIP subfamily. XIP proteins are located at the plasma membrane of the epidermal and parenchyma cells and at the endomembrane system. They transport certain polar solutes and water for specific members [30,33,34]. In filamentous fungi, and in a great majority of cases, the fungal XIP subfamily consists of a single member per species [35]. In *Trichoderma harzianum*, *XIP* is constitutively transcribed and up-regulated during interaction with the plant pathogen *Fusarium solani* on olive tree roots [32]. In silico analysis showed that fungal XIP transports water and possibly other small polar molecules like H_2_O_2_, but intriguingly, not glycerol concerning the plant counterparts. However, unlike some AQPs or AQGPs from fungi that have been functionally characterized [21,23,25,36], knowledge of XIP 3D structure, expression and physiological roles is still hypothetical. Thus, a fungal XIP prototype needs to be considered carefully.

In this context, the present study has a triple objective. After having introduced the MIP diversity in the *Trichoderma atroviride* genome, we first investigate the sequence and structural characteristic of the fungal XIP subfamily with a special focus on *Triat*XIP. Secondly, because the functions of the fungal-XIP are yet to be understood, the developmental, anatomical, physiological and biochemical effects of five XIP deletion mutants (∆*Triat*XIP) were subsequently analyzed under specific nutritional contexts and mycoparasitism situations against different phytopathogen agents. Thirdly, the XIP member must be considered as a unit integrating a network of channels, and profiling of the expression of the six other MIP paralogues present in the *T. atroviride* genome has been recorded in the deviant mutant phenotypes subjected to different contrasting abiotic environments. We make use of a comprehensive and multidisciplinary analytical approach, together with the results of recently published studies, to better understand the roles that the fungal-XIPs play in water transport and, more broadly, in the *Trichoderma*’ life cycle and its subjacent metabolisms.

## 2. Materials and Methods

### 2.1. Fungal Strains and Culture

*Trichoderma atroviride* strain IMI 206040 (wild type) was used for this study. The plant pathogenic strains *Botrytis cinerea* B05.10, *Fusarium graminearum* ACCC 36966, and *Rhizoctonia solani* ACCC 36246 were used as fungal preys. Strains were routinely inoculated on potato dextrose agar (PDA; peeled potatoes, 200 g; dextrose, 10 g; distilled water, 1000 mL; pH 6.5) in Petri plates, and incubated in continuous darkness at 23 °C. Chlamydospores were observed after 20 days of incubation on PDA in continuous darkness at 23 °C, and after 14 days on two corn flour media: cornmeal and water (200 g/L), pH 4.2, and corn flour 62.86 g/L, glycerol 7.54 mL/L, pH 4.2 [37].

### 2.2. Sequence Collection

The nucleotide and corresponding amino acid sequences of *MIP* genes were retrieved from *Trichoderma* spp. available at the Joint Genome Institute (JGI; http://genome.jgi-psf.org/ (accessed on 23 February 2021)) and the NCBI GenBank GSS databases (http://www.ncbi.nlm.nih.gov/ (accessed on 23 February 2021)). Each putative MIP sequence was carefully scrutinized including verification of the expected MIP motifs and the prediction of the transmembrane topology with Interproscan from EMBL (http://www.ebi.ac.uk/Tools/pfa/iprscan/ (accessed on 23 February 2021)). Protein names and accession numbers used in this work are listed in (Appendix A).

### 2.3. Precision on MIP Naming Used in this Work

In the bibliography and for convenience, the AQP and MIP terminologies (as well as their acronyms) are considered synonymous and are used interchangeably [38]. Because the MIP superfamily from fungi includes “true” AQP with strict water channel ability (which corresponds to a strict functional terminology), we use the acronym MIP for this study in a broader sense to describe the “aquaporin” channels *lato sensu*, and the aquaporin and its acronym AQP according to its strict functional terminology.

### 2.4. In Silico Molecular Characterization of Fungal XIP

Multiple alignments of the XIP protein sequences were computed with ClustalW, and the results were viewed with Jalview (v2.10.5). The phylogenic study was carried out with the coding region of *XIP* sequences. Sequences were aligned with MUSCLE (v3.8.31) [39]. Ambiguous regions (i.e., containing gaps and/or poor alignment) were removed with Gblocks [40]. The phylogenetic tree was reconstructed using the maximum likelihood method implemented in the PhyML program (v3.1/3.0 aLRT) [41]. Reliability for the internal branch was assessed using the bootstrapping method including 1000 bootstrap replicates. Graphical representation and edition of the phylogenetic tree were performed with TreeDyn [42]. MIP from *T. atroviride* (3 AQP: AQP-31598, AQP-43816, AQP-6990; 3 AQGP: Fsp-like_39327, Fsp-like_283564, AQGP-Other_90169; [28] were added to the analysis as XIP intraspecific out-group sequences, and the sequences from *Aspergillus terreus* (AAJN01000055.1) and *Penicillium marneffei* (XP002149425.1) were used as XIP interspecific out-group references [29]. The subcellular localization of *T. atroviride* MIP was predicted using WoLF PSORT, a protein subcellular localization prediction tool available at http://www.genscript.com/wolf-psort.html (accessed on 23 February 2021). In addition, predicted localizations were confirmed with the use of DeepLoc-1.0 (http://www.cbs.dtu.dk/services/DeepLoc/ (accessed on 23 February 2021)) and the Cello prediction system (http://cello.life.nctu.edu.tw/ (accessed on 23 February 2021)). Because these three tools generated similar data, only those using WoLF PSORT were shown in this work.

### 2.5. Homology Modeling and Atomic System Building

Homology molecular modeling of *Triat*XIP from *T. atroviride* was performed with the I-TASSER (Iterative Threading ASSEmbly Refinement) program suite [43]. As for the molecular dynamics simulations, the atomic system was built up using the charmm-gui web interface [44]. The homology model of *Triat*XIP was inserted into a 3-palmitoleoyl-2-oleoyl-d-glycero-1-phosphatidylcholine (YOPC) lipids bilayer and solvated with TIP3 water. An isotonic concentration of 150 mM KCl was then generated. The all-atom simulation was performed with Gromacs (v.2018.1) [45] in a charmm36 m force field [46]. A first minimization step was followed by six equilibration steps before the 10 ns production phase. Pressure and temperature were kept constant at 1 bar and 303.15 Kelvin, respectively, using the Berendsen method during equilibration and the Parrinello–Rahman and Nose–Hoover methods during production. The Lennard–Jones interactions threshold was set at 12 ångströms and the long-range electrostatic interactions were calculated according to the particle mesh Ewald method.

Concerning the trajectory analyses, permeability coefficients (*pf*) were calculated according to the collective coordinate method [47,48] from a 5 ns sub-trajectory after 5 ns of additional equilibration for each protomer leading to four repetitions. Water molecules were monitored through the MDAnalysis library [49]. The pore diameter was calculated using HOLE software [50] from five frames per nanosecond of the same sub-trajectory for each protomer.

For the structure analyses, electrostatic potentials were calculated with APBS [51]. Prior to this operation, PDB2PQR [52] was used with the PARSE forcefield [53] to type the atoms of the structure for the solver. PyMOL was used to analyze and illustrate the models [54].

### 2.6. Construction of the Gene Deletion Vector, Transformation and ∆TriatXIP Mutant Validation

Genomic DNA was isolated using a hexadecyltrimethylammonium bromide-based method [55]. The 5′-flank region (1.05 kb before the start codon) and 3′-flank region (including the stop codon) of *TriatXIP* were amplified from genomic DNA of *T. atroviride* using gene-specific primer pairs (Appendix A), and ups F/ups R and ds F/ds R as described previously [56,57,58]. Three fragment multisite gateway cloning systems (Invitrogen, Carlsbad, CA, USA) were used to construct gene deletion vectors. Gateway entry clones of the purified 5′-flank and 3′-flank PCR fragments were generated by BP reaction following the procedure described by the manufacturer (Invitrogen, Carlsbad, CA, USA). The entry clone for hygromycin resistance cassette (hygB) generated during our previous studies [56,57,58] from pCT74 vector [59] was used. The gateway LR recombination reaction was performed using the entry plasmid of 5′-flank, 3′-flank, hygB and destination vector pPm43GW [60] to generate the deletion vector. The deletion vector was transformed into *Agrobacterium tumefaciens* strain AGL-1 following a freeze–thaw procedure [61] and positive clones were selected on YEP (10 g/l yeast extract, 10 g/l bacto peptone, 5 g/l NaCl; pH 7.0) plates containing 35 µg/mL rifampicin (Sigma-Aldrich, St. Louis, MO, USA) and 100 µg/mL spectinomycin (Sigma-Aldrich, St. Louis, MO, USA).

*Agrobacterium tumefaciens*-mediated transformation (ATMT) was performed based on a previous protocol for *T. harzianum* [62]. Transformed strains were selected on plates containing 100 µg/mL of hygromycin (Sigma-Aldrich, St. Louis, MO, USA). Validation of homologous integration of the deletion cassettes in putative transformants was carried out using a PCR screening approach with primer combinations targeting the hygB cassette and sequences flanking the deletion cassettes as described previously [56,57,58]. The PCR-positive transformants were tested for mitotic stability and purified by two rounds of single spore isolation [56,57,58]. The transcript level of the *TriatXIP* gene in *T. atroviride* wild type and gene deletion strains were determined by RT-PCR using RevertAid premium reverse transcriptase (Fermentas, St. Leon-Rot, Germany) and their respective primer pairs (Appendix A). For all analysis, five independent ∆*TriatXIP* deletion strains (named ∆*TriatXIPa* to ∆*TriatXIPe)* were used to confirm that the observed phenotypes were attributed to the deletion of *TriatXIP* but not to ectopic insertions.

### 2.7. Histological Analysis

The morphology of each strain was recorded by optical microscopy (×40 and ×100). After 10 days of cultivation, mycelium was absorbed on clear tape, stained with 1% (*w*/*v*) toluidine blue for 1 min, washed with water for 5 s, and then apposed on microscope slides. All images were processed with a Zeiss Axio Observer Z1 microscope, digital camera and Zen imaging software system (Zeiss, Jena, Germany).

### 2.8. Screening for Antagonistic Activity of T. atroviride

In vitro antagonistic activity of the *T. atroviride* strains (wild type and the five ∆*TriatXIP* strains) were independently evaluated against three phytopathogenic fungi: *B. cinerea*, *F. graminearum*, and *R. solani*. A 5-mm-diameter disc of actively growing mycelia of the *T. atroviride* strains was placed on the periphery of a PDA plate. Similarly, phytopathogens were placed on the opposite side of the plate. Assays were incubated at 23 °C for 10 days. Antagonistic activity was measured as zone inhibition and growth reduction for a period of 10 days. All experiments were conducted in independent biological triplicates.

### 2.9. Biolog Phenotype Microarray Experiments

Phenotype MicroArrays™ for Filamentous Fungi (Biolog FF, Biolog Inc., Hayward, CA, USA) were used to compare the metabolic profiles on 95 single compounds of the *T. atroviride* wild type strain and the five ∆*TriatXIPs*. A list of the compounds and their biochemical families is given in Appendix A. The assimilation of compounds was reflected after 96 h of mycelial growth and quantified by measuring the optical density (OD) in the wells at 750 nm, the wavelength at which hyaline mycelium has maximum absorbance. To quantify catabolism, the wells contained a tetrazolium dye that turns into a purple insoluble precipitate when reduced due to mitochondrial activity. This was measured at the maximum absorbance of the reduced tetrazolium salt of 490 nm. Beforehand, conidia of each strain were generated on PDA plates at 23 °C in constant darkness. The FF plate procedure was carried out essentially as per the manufacturer’s instructions, with slight modifications [63,64]. The experiment was performed on three replicate plates with separately prepared inocula, and incubated at 23 °C under constant darkness.

The statistical analysis and graphical output were performed using R software (version 3.6.3, R-core Team) complemented by FactoMineR (2.3) [65], FactoExtra (1.0.6) [66], tidyverse (1.3.0) [67], ggplot2 (3.3.0) [68], gridExtra (2.3) [69], egg (0.4.5) [70], lemon (0.4.3) [71], multcomp (1.4.12) [72], multcompView (0.1.8) [73], broom (0.5.5) [74] and RcolorBrewer (1.1.2) [75] libraries. Data was scaled to unit variance for PCA. Data smoothing of Appendix A was obtained after a linear model fit. Multiple comparisons of Appendix A were made under the generalized linear model with data fitted to a single stratum analysis of variance and the Tukey honest significant difference test. *p-*values were rounded to the third decimal place.

### 2.10. Screening for MIP Transcriptional Responses of T. atroviride in Various Nitrogen and Carbon Amended Media

A 5 mm disc of the wild type and the five ∆*TriatXIP* strains were inoculated on PDA medium amended with 0.5 M of D-Glucose, D-Ribose, D-Mannose, NO_3_NH_4_, Urea, Glycerol, D-Mannitol, L-Asparagine and L-Aspartic acid, and incubated at 23 °C under constant darkness for five days. At the end of the experiment, the mycelia were removed from the plates and stored at −80 °C for further molecular analysis. All experiments were conducted in independent biological triplicates.

### 2.11. RNA Extraction and Quantitative RT-PCR

Total RNA was extracted from mycelia after being ground to a fine powder in liquid nitrogen and treated with a CTAB extraction buffer (Bromide Cetyltrimethylammonium) as previously described [28]. RNA concentrations were determined by spectrophotometry at OD 260/280 (spectrophotometer ND-1000, Nanodrop, France), and quality was checked by using 2% TAE/agarose electrophoresis. Two µg of total RNA were reverse-transcribed with Oligo-dT using the SuperScript^®^ III First-Strand Synthesis System for RT-PCR (Invitrogen, Carlsbad, CA, USA). cDNA was diluted 40-fold with sterile water. The abundance of MIP-related transcripts was determined by real-time qPCR with a MyiQ instrument (Bio-Rad, Hercules, CA, USA). MIP gene expression levels were calculated by the 2^-∆∆*C*T^ method [76]. PCR amplifications were done in 15 µL of PCR reaction using MESA GREEN qPCR MasterMix Plus (Eurogentec, Liège, Belgium) from 2 µL of 40-fold diluted cDNA template. PCR cycling conditions were 1 cycle for 3 min at 95 °C, followed by 35 cycles for 15 s at 95 °C, 15 s at 54 °C, and 20 s at 72 °C. PCR reactions were ended by generating a dissociation curve of 56 to 94 °C with an increment of 0.5 °C/10 s to check for primer dimers and nonspecific amplification. All PCR technical samples were assayed in duplicate. The geometric mean of *C*_t_ of five genes encoding to *18SrRNA*, *28SrRNA*, *Tubulin*, *Actin*, and *Gpdh* were used as internal references to normalize MIP expression for their stable expressions during fungus treatment. These genes were chosen from a panel of widely used housekeeping genes [28,29,30,31,32,33,34,35,36,37,38,39,40,41,42,43,44,45,46,47,48,49,50,51,52,53,54,55,56,57,58,59,60,61,62,63,64,65,66,67,68,69,70,71,72,73,74,75,76,77] using the software application BestKeeper v1 [78]. Each referrer belongs to protein families involved in different cell processes in order to minimize the risk of co-regulations. Specific primer pairs for each MIP member were designated in consensus zones after the alignment of MIP sequences retrieved from *Trichoderma* spp. with the Primer3plus application (http://www.bioinformatics.nl/primer3plus (accessed on 23 February 2021)). Specific amplification of only one desired band was observed using each primer combination for qRT-PCR analysis, and PCR efficiency was 100 ± 3% for all primer pairs. Primer pairs are listed in (Appendix A). For statistical analysis, the biological replicates correspond to the five *TriatXIP* null mutants and three wild type independent mycelial cultures. Gene expression levels between parental and mutant strains were computed by a one-way analysis of variance (ANOVA) followed by a Tukey’s honest significant difference (HSD) post hoc test (*p* < 0.05).

## 3. Results and Discussion

### 3.1. Tichoderma atroviride MIPsub-class

#### 3.1.1. *T. atroviride* XIP, AQP and AQGP Inventory

In recent years, the physiological relevance of the MIP-facilitated transmembrane diffusion of water and various solutes has been largely acknowledged in Eukaryotes such as plants and animals. Comparatively, such information from fungi is lacking, except for ectomycorrhizal (EcM) fungi that develop mutualistic associations with the roots of a large range of plants. Fungi exhibit great phenotypic plasticity in their responses to their immediate growth environment, correlated with remarkable nutritional versatility that models a great variety of polyphagical lifestyles. MIP integrates a wide and complex network of transmembrane channels that provides controlled cell internalization of inorganic and organic matter from the immediate environment. This matter, regarded as the molecular building blocks of life, is of prime importance in the context of nutrition (metabolism) as well as for maintaining cell homeostasis, osmoregulation and turgor, and determining the final shape of the cells. The MIP superfamily from the *Trichoderma* genus includes the fungal aquaporins (AQPs) (*stricto sensu*) with the “orthodox fungal water channels”, the fungal aquaglyceroporins (AQGPs) with the *Fps*-like aquaglyceroporins and the facultative fungal aquaporins (or “other” aquaglyceroporins), and the presumed fungal (un-)orthodox aquaporins *X*-intrinsic protein (XIP) (Figure 1 and Appendix A) [28,29,30,31,32,33,34,35,36,37,38,39,40,41,42,43,44,45,46,47,48,49,50,51,52,53,54,55,56,57,58,59,60,61,62,63,64,65,66,67,68,69,70,71,72,73,74,75,76,77,78,79]. Concerning *T. atroviride*, its genome contains seven *MIP* homologous genes that cover the three MIP sub-families: three *AQGPs* including one “*other AQGP*” (JGI Protein id: 90169; then called *Triat*OtherAQGP-90169) and two *Fps-like AQGPs* (*Triat*FpsAQGP-39327, *Triat*FpsAQGP-283564), three *classic AQPs* (*Triat*AQP-31598, *Triat*AQP-43816, *Triat*AQP-6990), and one *XIP* (*Triat*XIP-319992) (Appendix A). Every related protein falls under the MIP archetype: each member is a *small* integral protein, ranging from 276 to 344 AA, with theoretical MWs from 30.132 to 37.57 kDa and *p*Is from 6.60 to 9.14, except for *Triat*FpsAQGP-283564 with 594 AA, 65.625 KDa, and a *p*I 5.88, respectively; Table 1), for which every 3D prediction is confidently an aquaporin structure (Appendix A).

In this respect, their 3D structure resembles that of an hourglass formed by six transmembranes α-helices (TMH1 to TMH6) connected by five inter-helical loops (Loop A to E). The *N*- and *C*-termini are localized intracellularly. At the center of the pore formed by the six TMH helices, two distinct constriction structures are formed: one helicoidal with two highly conserved “NPA” (Asn-Pro-Ala) motifs, and the second called “ar/R” selectivity filter (ar/R SF). The latter is composed of four specific amino acid residues that spread along the MIP primary structure but whose side chains fit into the channel (Table 1). These two constrictions act as a selective filter, determining the solute permeability of the MIPs [80]. Finally, the in silico prediction of the spatial expression of *T. atroviride* MIPs shows that they are predominately found in plasma membranes, but may also be located in endoplasmic reticulum, peroxisomes, mitochondria and vacuoles (Appendix A).

MIPs in pathogenic fungi may act as attractive targets for antifungal drugs [35]. In this respect, knowledge emerges on certain fungal AQP and AQGP counterparts, which is not the case for the “uncharacterized *X*-intrinsic protein” subgroup. The fungal *XIP*, with *Triat*XIP (protein ID 319992) as a representative member, is the core of our issue for this present work.

#### 3.1.2. Sequence and Structural Characteristic of TriatXIP

In light of the available sequenced genomes from fungi, and coinciding with what is commonly observed in the genomes of several filamentous fungi [28,29,30,31,32,33,34,35,36,37,38,39,40,41,42,43,44,45,46,47,48,49,50,51,52,53,54,55,56,57,58,59,60,61,62,63,64,65,66,67,68,69,70,71,72,73,74,75,76,77,78,79], one XIP gene is present in the *T. atroviride* genome (named *TriatXIP*). *TriatXIP* is made up of four introns and six exons, resulting in a CDS of 906 nucleotides of length. *TriatXIP* is predicted to encode a protein of 301 amino acids, named *Triat*XIP, with a theoretical molecular mass of about 32 kDa which typically corresponds to an MIP protomer. Like any typical MIP, the *Triat*XIP protein protomeric structure is highly conserved and resembles an “hourglass model” (Appendix A). It is composed of six transmembrane alpha-helix (TMH1-TMH6) connected by five extramembrane loops (LA-LE). The central pore comprises two additional and short helical segments (HB and HE) which both possess a highly conserved “Asn-Pro-Ala” (NPA) motif that forms a narrow region of the pore (NPT on HB and NPA on HE in *Triat*XIP). Because of the positive electrostatic field generated locally by the hemi-helix dipole, this region is believed to be essential for proton exclusion [81]. The substrate selectivity of MIP mainly depends on another constriction: the aromatic/arginine (ar/R) selectivity filter composed of four residues [82,83,84,85].

In order to access the atomic details of water transport through *T. atroviride* XIP, a homology model of its three-dimensional structure was built following the same process as in our previous work [28]. However, this time, the model was integrated into an atomic system mimicking the cellular conditions encountered by this transmembrane protein family (i.e., inserted in a lipidic bilayer and solvated with water and 150 mM KCl ions) and molecular dynamics were performed.

The ar/R constriction residues of the *Triat*XIP are displayed in Figure 2a, Appendix A and Table 1. The first three residues, N81, S211 and Q225 are typical of XIP ar/R constriction and correlate with previously observed data on *T. harzianum* [28]. However, the conserved Arginine (R) is replaced by a Lysine (K) on *T. atroviride* XIP. While the Lysine side chain is also positively charged, its interactions with water will not be exactly the same as the Arginine guanidinium group, and could change the transport properties of the pore. The surface electrostatic potential of the pore shows the expected radiation of the ‘ar/R Arginine or Lysine’ side chain positive charge in the upper part of the channel (Figure 2b) as well as the positive electrostatic field located at the center of the pore and generated by hemi-helices HB and HE dipoles. Finally, we can also observe the local negative regions created by the oxygens of pore lining residues which correspond to protein-water interaction sites (Figure 2b). The mean (± standard error) pore diameter for *T. atroviride* at the ar/R constriction is 1.86 ± 0.1 Ångströms which is close to the *T. harzianum* XIP pore diameter at this same location of 1.8 Ångströms [28]. This substitution by Lysine seems to be exclusive to these few fungal-XIP [35]. It is not observed in MIPs from plants, animals, insects or Prokaryotes, where Arginine is replaced by Asparagine or more marginally, inter alia, by Valine, Cysteine or Serine [29,30,86]. It will be very interesting to understand what kind of functional effect this residue change might have on the protein and, ultimately, on the fungal behavior.

We can also notice the absence of the salt bridge on the HE hemi-helix which is thought to allow for a small shift of the helix position and, hence, to increase pore size in AQGPs [28]. The conserved Aspartate located one position after the ‘ar/R Arginine’ in AQGPs is replaced by a Cysteine in *T. atroviride* XIP, however this Aspartate and the Arginine situated one helix turn after are conserved in the three AQGPs of *T. atroviride* (Appendix A).

From the 5 ns of sub-trajectory simulation produced and sampled from a 100 ns full simulation result [47], we can observe water permeation as well as a typical single file organization of water molecules inside the pore, especially in the NPA region, the vestibules looking a little more disorganized than in strict AQPs (see movie Appendix A). Using the collective coordinate method [47,48], water permeability coefficients (*pf*) were calculated for each protomer. The mean (± standard error) *pf* is 2.7 ± 0.4 × 10^−14^ cm^3^s^−1^, which is typical for *pf* values of AQPs computed from molecular dynamics [87,88,89].

To conclude, based on the similar electrostatic profiles of strict AQPs and *Triat*XIP pores and the similar organization of water continuum and permeability coefficients, it seems very likely that *Triat*XIP is an efficient water channel. However, the wider ar/R constriction and the replacement of its typical Arginine by Lysine might generate functional differences between *Triat*XIP and strict AQPs, and might favour small polar solutes other than water.

### 3.2. Functional Characteristics of T. atroviride TriatXIP

#### 3.2.1. General Issue and Objectives of This Item

There is an increasing understanding that the MIP superfamily performs an astonishing variety of physiological functions that define the biological activities of organisms. Among the fungal MIPs, XIPs are not functionally characterized to date, which means their role is still obscure. Promising channel functions predicted by structure modeling reveal that the plasma membrane *Triat*XIP proteins might act as strictly aquaporin-like proteins. However, we cannot exclude that *Triat*XIP proteins can transport in vivo specific solutes as some orthodox aquaporins and/or aquaglyceroporins, possibly due to their ability to interact with potential regulators.

Furthermore*, TriatXIP* is part of the five *TriatMIP* genes that are transcripted in *T. atroviride* (this item will be discussed in greater detail later in this article). It is hazardous to correlate the steady state levels of the transcriptional expression of a gene with the putative functions of its related protein; however, it is very attractive to consider the XIP channel among the potential candidates facilitating the diffusion of matter across the plasma membrane involved in fungal metabolism and growth. Accordingly, it could be expected (but nonetheless questionable) that the lack of a single aquaporin belonging to a superfamily directly impairs a wild phenotype—in this case, *T. atroviride*.

Indeed, should *TriatXIP* be significantly expressed and predicted to transport water, its failure among the fungal MIP network may impact the overall intracellular water content and intracellular turgor pressure. Such a hydric upheaval, even though infinitesimal, can reset the overall metabolism of the organism. To help shed light on some potential XIP roles in fungi, five *TriatXIP* deletion mutants (∆*TriatXIP*) from *T. atroviride* were designed, and their phenotypes, cell metabolism, and subjacent MIP transcriptional profiles were analyzed in strains developing in stressful biological conditions.

#### 3.2.2. *Triat*XIP Disruption Moderately Impacts Fungal Growth but Annihilates Chlamydospore Formation

Despite the transcript accumulations of *TriatXIP* in the wild type, *TriatXIP* deletion does not impact fungal growth during the first 48 h of cultivation under stress-free growing conditions (Figure 3a). Contrasted mycelial growths in ∆*TriatXIP* are significantly distinguishable from the third day of cultivation. Ultimately, all *Trichoderma* strains intensively cover the entire surface of the Petri dishes after four days of cultivation for the wild strain and five days for the transformants. This is consistent with studies carried out on yeast or filamentous organisms where nullified *AQP* or *AQGP* gene expressions result in growth failure and are connected to an assumed decrease in plasma-membrane water permeability [23,90,91]. In this context, considering the pivotal role of water channels in maintaining cellular homeostasis and metabolism integrity, *TriatXIP* disruption may result in osmoregulatory imbalance that significantly impacts *T. atroviride* growth.

At a microscopic level, subtle perceptible differences in the hyphal morphology of transformants compared to the wild type strain are revealed (Figure 3b and Appendix A). Concerning the conidiospores, their appearance is similar between strains. The most notable difference is the very low presence of chlamydospores (or even a complete lack according to the experiment; Appendix A) with the mutants after 20 days of cultivation on a PDA medium and specific corn flour media [92]. Furthermore, most chlamydospores display abnormal morphologies and/or are somewhat smaller (≈6.5 µm) than those observed with the wild strain (≈8.5 µm) (Figure 3c). These data would indicate that *TriatXIP* plays a role in the chlamydosporogenesis in *T. atroviride*, thus directly interfering with one of the asexual reproduction processes of the fungi. Chlamydospores usually form from the mycelium or certain conidia and play a major role as survival structures for various pathogenic and saprophytes fungi. However, even today, very little is known about the molecular networks involved in the formation and maturation of chlamydospores and the natural factors that enhance their formation, especially for the *Trichoderma* genus [92,93]. A genome-wide transcriptional analysis of these mutants will offer an opportunity to further understand the chlamydosporogenesis process in *Trichoderma*.

#### 3.2.3. Evaluation of *T. atroviride* Wild Type and ∆TriatXIP Mutants as Mycoparasites

A great deal of literature describes *T. atroviride* as a mycoparasite, drastically reducing the mycelial growth and conidiation of a panel of economically important aerial and soil-borne phytopathogenic fungi [94]. In this respect, this strain is now increasingly used as a bioprotectant in organic cropping systems around the world. *Trichoderma* biocontrol properties are described as a complex combination of several mechanisms, including nutrient competition and direct mycoparasitism, which involves the production of antifungal metabolites and cell-wall degrading enzymes. Although very little is known about the contribution of MIPs in establishing and deploying mycoparasitic mechanisms of biocontrol agents [91,92,93,94,95], it has recently been suggested that aquaporins are involved in the *Trichoderma*-prey interactions [28,95].

To substantiate the role of *TriatXIP* during antagonism, *T. atroviride* wild type and the five ∆*TriatXIPs* were confronted with *B. cinerea*, *R. solani*, and *F. graminearum*, given that *T. atroviride* stops the growth of these phytopathogens. The results show that, despite a significant slowdown in mycelia growth when wild type, transformants and prey mycelia meet together at four days of cultivation; no specific knock-on effect in growth kinetics for the ∆*TriatXIPs* is observed throughout the confrontation (i.e., before and until the full draping of the mycelia) (Figure 4 and Appendix A). Similarly, no differences in growth kinetics of the fungal prey fungi are recorded during the first steps of interaction with the ∆*TriatXIPs* in comparison with the wild type, nor after six days of dual confrontation. When contact is established between strains, prey growth is definitively stopped. The data clearly show that ∆*TriatXIPs* are capable of annihilating phytopathogen growth as with the wild type. This demonstrates that deleting *TriatXIP* does not cause the apparent failure in transformants in an antagonistic capacity, nor does it alter their ability to confront the potential aggressivity of the preys, at least with the phytopathogen strains used.

#### 3.2.4. Metabolic Analysis of ∆*Triat*XIP Mutants

The XIP loss of function in ∆*Triat*XIP mutants slightly delays biomass accumulation. At the microscopic level, these mutants express discrete differential phenotypic traits on the hyphae and nearly zero chlamydosporogenesis generating mostly chlamydospores with anomalous morphologies. In addition to the fact that fungal cells must continuously regulate the water permeability of their membranes in order to sustain water uptake during growth and/or osmotic adjustment [20], a phenomena in which *Triat*XIP may play a role; it is possible to hypothesize that *TriatXIP* can play a wider role by regulating *C*- and *N*-matter flow, notably in metabolic homeostasis. To answer this question, we analyzed the metabolic profiles of the ∆*TriatXIPs* in comparison with the wild type strain. To do so, we placed these analyses in a larger context where, in a natural composting environment, the microorganisms are capable of metabolizing and competing for a high variety of vital compounds (or macronutrients). Filamentous fungi, such as the *Trichoderma* genus, play a vital role in the ecological equilibria between microorganisms, in particular by being responsible for a significant proportion of the hydrolysis of soil resources [96]. Considering that macronutrients are one of the major determinants of fungal phenotype, we used the high-throughput method of the Phenotype MicroArray™ (PM) system (Biolog FF) to explain phenotypic changes of the ∆*TriatXIP* strains to contrasted nutrient utilizations. Based on the absorbance measurements at OD_750_, which reflect biomass production (i.e., mycelial growth), and OD_490_, which reflects the respiration rate and the substrate use (i.e., catabolism), this phenotype microarray assay has been introduced as a way to do high-throughput screening of diverse basic nutrient sources as well as evaluate growth and metabolic characterizations of various microorganism strains, including assorted *Trichoderma* spp. [14,15,16,17,18,19,20,21,22,23,24,25,26,27,28,29,30,31,32,33,34,35,36,37,38,39,40,41,42,43,44,45,46,47,48,49,50,51,52,53,54,55,56,57,58,59,60,61,62,63,64,65,66,67,68,69,70,71,72,73,74,75,76,77,78,79,80,81,82,83,84,85,86,87,88,89,90,91,92,93,94,95,96,97]. However, the interpretation of each between-strain difference in the speed of substrate use is not straightforward and has to be done with great care: each change in color could be produced by a redox shift in tetrazolium dye due to cell respiration (NADH production) that can interplay with (i) inherent variations in some enzymatic activities of the fungi during its growth, (ii) an increase of turbidity due to fungal body proliferation, or (iii) a change in medium color after producing specific metabolites in response to the candidate metabolite of the microplate and/or by the production of metabolic wastes by the fungi. Nevertheless, and despite the methodological uncertainties, a change in absorbance values can be interpreted as a useful indicator of metabolic activities that can suggest general metabolic trends. In this respect, this approach has proved effective by offering many cell-wide perspectives, especially in understanding how a gene initially encoded in the genome and under control of the environment is displayed in fine at the cellular level, notably throughout the phenotypes of an organism [98,99].

In the present work, the metabolic profiles tied to 95 *C*- and *N*-compounds were analyzed (metabolites listed in Appendix A). As observed for several fungal strains (e.g., *Acremonium*, *Aspergillus*, *Neurospora*, and so on), and in line with the metabolic profiles recorded for species from the genera *Trichoderma* [100], *T. atroviride* exhibits a wide range of substrate assimilation (respiration) and rapid growth (biomass) (both being significantly colinear—Appendix A) which covers the major biochemical classes (Figure 5 and Appendix A). Such physiological aptitudes can be brought into correlation with the great advantage for this genus of colonizing many worldwide ecological niches. In detail, the most assimilated carbon sources are carbohydrates (oses and osides) followed by polyols, carboxylic acids, and amino acids. The nucleic acids and the amino- and amido-compounds are consumed at a much lower level. Still to date, the metabolic richness structuring the functional diversity of microbial communities (in soil and/or within a “host” organism) are far from being listed.

First of all, this data integrates the paradigm that carbohydrates contribute most to the overall metabolic activity of microbiota, and that the related carbohydrate regulatory pathways appear to be modulated mainly through perturbations in the communities [101,102].

#### 3.2.5. Comparative Analysis of Transformants and Parental Strains

Concerning the comparative analysis of cell responses between the transformants and the parental strain, 16 differential metabolic patterns can be significantly identified (*p* < 0.05) (Figure 5 and Appendix A). There is decreased metabolization in seven oses, five amino acids, three carboxylic acids and one polyol*,* reflecting a possible reduction in the resource uses.

Concerning the oses, disruptant phenotypes are perturbed on d-Mannose (*p* < 0.0001), d-Ribose (*p* = 0.007), N-acetyl-d-Glucosamine (*p* = 0.013), β-Methyl-d-Galactoside (*p* = 0.013), Glycerol (*p* = 0.015), and d-Arabinose (*p* = 0.03). In fungi, Mannose is an interesting candidate because it integrates multifaceted pivotal functions in cell physiology. For example, it is notably present in the composition of a panel of mannose polymers, mannans and soluble peptidomannans, which form an integral part of the cell wall surface of most fungi [103]. The Mannose residues here are crucial for maintaining the cell wall integrity and fungal viability [104]. Furthermore, Mannose is considered potent cell organic protectant, as it may rank among the most abundant polyols (with trehalose and raffinose) in cells exposed to various stress conditions. In this respect, it can constitute up to 10% of the dry weight of certain filamentous fungi [105]. As regards the nutritional context, the Mannose polymers that structure the fungal cell wall surface or those that are in the compositions of the plant cell wall pectic polysaccharides (rhamnogalacturonans) and several plant secondary metabolites (e.g., anthocyanins, flavonoids and triterpenoids) can be metabolized. The releasing d-Mannose can be used as sole carbon sources, then being metabolized to supply various fungal pathways including the pentose-phosphate cycle, glycolysis and the amino sugar and nucleotide sugar metabolisms. Similarly, N-acetylglucosamine (GlcNAc), an amide derivative of the monosaccharide glucose, is the major monomer of the polysaccharide chitin (β-(1,4)GlcNAc). Chitin is an essential structural component located in the inner layer of the fungal cell wall close to the plasma membrane. With various *β*-(1,3/1,6)-glucans, they constitute the structural scaffold of the cell wall, thus conferring strength and rigidity to the cell wall as a whole in most fungi [106]. Concerning β-Methyl-d-Galactoside, a methyl derivative of the monosaccharide galactose, it integrates the composition of various galacto-containing oligosaccharides and branched polysaccharides such as galactomannans and galactoglucomannans. These polymers are the main groups of plant hemicelluloses, but they have also been isolated from cell walls in several fungi. Interestingly, these two monomers are linked to Mannose metabolism through the amino sugar and nucleotide sugar metabolisms. d-Ribose and d-Arabinose (two major constituents of the plant cell wall lignocelluloses) are two of the intermediates of the pentose catabolic pathway, and in this respect, several fungi can use these aldopentoses as their sole carbon sources. In *Trichoderma*, the pentose sugar pathway involves several enzymatic steps, most of which have been well-characterized, especially those linked to the applications of certain sugars in food and the feed and biofuel industries [107,108,109]. Fundamentally, it generates NADPH and pentoses, some precursors that are of great importance for the biosynthesis of nucleic acids and amino acids. Interestingly, the mutants show a decrease in metabolic activity when they develop on the pentose pathway intermediates d-Arabinose and d-Ribose (as with d-Mannose and its derivative L-Rhamnose). It is plausible that the pentose catabolic pathway is affected in the transformants, impacting de facto their metabolic activities and redox equilibrium. It may explain the subtle changes during fungal growth, as previously observed for *Candida albicans* [110]. Among the polyoses, interesting statistical trends emerge involving reducing diholosides d-Cellobiose (*β*-d-Glc*p*-1,4-d-Glc*p*) and d-Melibiose (α-d-Gal*p*-1,6-d-Glc*p*) which noticeably alter ∆*TriatXIP* growth and respiration in comparison with the parental strain (*p =* 0.061, *p =* 0.087, resp.). These diholosides are produced through hydrolysis of the plant cell wall by fungi before being internalized to supply energy to cells.

When considering the group of substrates belonging to the amino acid class, and particularly those that produced the most significant biomass, significant metabolic differences can be found between wild type and transformants such as L-Alanyl-Glycine (*p* = 0.005), L-Aspartic acid (*p* = 0.013) and L-Asparagine (*p* = 0.032). When amino acid use induces very moderate biomass, differential growth is significantly observed on L-Phenylalanine (*p* < 0.0001) and Glycyl L-Glutamic acid (*p* = 0.006). Besides being proteinogenic *amino acids**,* several of these amino acids can be diverted to various metabolic pathways such as the Krebs–Henseleit TCA cycle, providing cell energy. For example, L-Asparagine and its derivative L-Aspartic acid are linked to the TCA intermediate deaminated keto acid oxaloacetate. This implies that these amino acids are preferred by *T. atroviride* to supplement the supply of mineral nitrogen and energy, as previously observed with the Basidiomycota *Scleroderma sinnamariense* [111]. However, the amino acids L-Serine, L-Alanine, L-Threonine and L-Succinic acid are metabolized as often as L-Asparagine without inducing contrasting biomass between strains (Appendix A). Consequently, a possible interpretation is that the nitrogen pathways would be impacted, particularly the urea cycle, as it is intricately linked to the TCA cycle, first and foremost. This hypothesis is strengthened by the fact that ∆*TriatXIP* biomass and respiration give an interesting decreased tendency on L-Ornithine (*p* = 0.078). However, the power of the experimental design (with three repetitions for each strain) hinders any clear decision. The urea cycle integrates biochemical reactions to convert ammonia (NH_3_), a toxic waste originating from the amino acid catabolism, into urea ((NH_2_)_2_CO), a less toxic nitrogen source for excretion. From there, under conditions of amino acid fluctuation, L-Asparagine and L-Aspartic acid, with their derivative oxaloacetate, are known for regulating various enzymes from the urea cycle, maintaining the flow of nitrogen into the urea cycle via the carbamoylation of the L-Ornithine by the Ornithine transcarbamoylase enzyme. The fact that XIP transporters can be linked to the urea cycle opens up a whole new realm of possibilities that should be explored in future experiments, especially since XIPs are known for facilitating the transport of urea [33]. Lastly, L-Asparagine is one of the alternative forms of storing nitrogen in plants, and consequently, could be a favorable source of organic nitrogen for fungal growth. Interestingly, several amino acids including L-Asparagine and L-Aspartic acid triggered fungal metabolic activity, notably by playing key roles in stimulating the expression of many genes, including those required for the production of extracellular hydrolases (i.e., proteases, chitinases and glucanases) and transporters [112]. These amino acids, as regulators, integrate the nutritional processes whatever the lifestyle of the fungi, parasitism or saprophytism [113].

To conclude, several of the *C*- and *N*-compounds mentioned above are quite common in yeast and fungi (including *Trichoderma* spp.). They contribute to many key physiological processes by modulating spore germination, (a-)sexual development and forming fruiting bodies, and enhancing bioactivity of vegetative cells. However, the interconnections between these metabolic pathways are eminently complex and warrant further discussion, especially in fungi.

Our data clearly demonstrate that the direct mutagenesis of *Triat*XIP alters central *C*- and *N*-metabolisms in *T. atroviride*. However, it is very tricky to correlate deviant metabolisms with metabolites that would not be transported by XIP a priori, due to the configuration of the canal, which is too narrow. This is all the more important since a plausible alternative mechanism deserves to be highlighted: it cannot exclude that the *Triat*XIP deletion has a direct impact on cell respiration by disrupting gas exchanges, in particular for O_2_ and/or CO_2_. Recent research has revealed that some human and plant MIPs are involved in the transport of these hydrophobic gazes [114,115,116]. Although the underlying functional and structural mechanisms are not yet clearly elucidated [117,118], this new evidence opens a very interesting horizon, especially for the XIP subclass, which presents several structural singularities and for which the transport of gaseous molecules has not yet been characterized. Clearly, more investigation is required to confirm how *Triat*XIP participates in the C- and *N*-metabolisms and energy-releasing pathways, and how this assembly incorporates Chlamydosporogenesis.

#### 3.2.6. TriatXIP Transcriptional Regulation in the Wild Strain

Irrespective of how ∆*TriatXIP* transformants grow, and even though the most significant variations in carbon and nitrogen source use profile is limited to no more than a few molecules, the fact is that a lack of *TriatXIP* interferes with both the carbohydrate and nitrogen pathways. Of the numerous cell functions that could be awarded to these metabolites, those for which they act as sensing compounds and/or growth regulators attract special attention. For example, d-Mannose, d-Ribose, d-Cellobiose and d-Melibiose are known to be potent natural inducer components that modulate the production and/or activity of various classes of hydrolases in *Trichoderma* spp. [108,119,120].

In this regard, because exchanges of matter between compartments are constantly and finely regulated, there is a strong likelihood that any fluctuating events in the immediate environment of the fungi will modulate the transcriptional steady-state levels of *TriatXIP*.

Consequently, for the transcriptional analysis, we used the assays inherent to some nutritional complementations with various nitrogen (urea, NH_4_NO_3_) or *C*- and *N*-elements (d-Mannose, d-glucose, d-Ribose, L-Aspartic, L-Asparagine), and two osmoprotectants polyols (Glycerol, Mannitol). In the wild type, results show that *TriatXIP* is significantly induced when PDA is complemented with NH_4_NO_3_, d-Mannose, L-Aspartic and L-Asparagine (Figure 6a). *A contrario*, *TriatXIP* is significantly repressed by d-Glucose and Mannitol. Glycerol, Urea and d-Ribose have no effect on *TriatXIP* expression. Scanning 1.5 kb upstream of the start codon of *TriatXIP* using the yeast *Saccharomyces cerevisiae* dedicated promoter database SCPD showed that these modulations can be attributable to an over-representation of *cis*-regulated motifs targeted by nucleic acid binding proteins known to be involved in various carbon and nitrogen metabolic processes (data not shown). These results support previous analyses conducted on the promoter of *XIP* from *T. harzianum* that exhibits similar *cis*-element patterns [28].

The sensing functions dependent on specific *C*- and *N*-metabolites are possible ways of cell regulation *Triat*XIP dependent. This hypothesis fuels very exciting fundamental questions. Because metabolisms are intricate, our data offer a new opportunity to explore the cellular signaling as non-linear pathways that need to be further studied in fungi, especially for the *Trichoderma* genus. However, fundamentally, the assimilation of the specific components mentioned above could be viewed through the multifunctional permeant-channel abilities of the members constituting this diversified MIP superfamily to which *Triat*XIP belongs.

### 3.3. The Fungal XIP, AQP and AQGP: Three Pivotal MIP Pawn Families on Complex Chessboard Membrane Transport

In the wild strain, the *TriatXIP* gene is co-transcribed with two AQP genes (*AQP-31598* and *AQP-43816*) and two *Fps-like* genes (*AQGP-39327* and *AQGP-*283564) (Figure 6b). The *AQP-43816* and the “*Other”-AQGP-*90169 do not seem to be expressed in our biological conditions. On the nutritional environments described above, the four *Triat*MIPs transcribed are up-regulated in most of the studied conditions, with the exception of AQP-31598 and AQGP-39327 which are repressed in the presence of d-Glucose and/or Mannitol. AQP-43816 and AQP-31598 are not modulated by Mannitol, as for AQGP-39327 with Urea. More importantly, these transcriptional patterns significantly contrasted with those monitored with the ∆*TriatXIP* mutants: the down-regulations of the four *Triat*MIPs expressed observed with the wild type are exacerbated with the transformants, except for AQP-43816 with Mannitol, and AQGP-283564 with d-Glucose and Mannitol, which are up-regulated (Figure 6b and Appendix A).

It is of great interest to assess the MIP gene expression in its entirety. Indeed, we recall (i) the representativeness of the three fungal MIP subgroups in the transcriptional network of the fungi with at least one member per subgroup which is expressed and modulated in fluctuating environments, (ii) the substantial differences in gene regulation between members from a same subgroup that share relative high sequence identity, and (iii) a subcellular localization of all members that is preferentially predicted to plasma membrane. In addition, despite the significant induced expression of *TriatXIP* in the wild type strains under contrasted environmental conditions, the ∆*TriatXIP* mutants show moderate altered phenotypes and seem to conserve their mycoparasitic features. In view of all these circumstances, simplistic and then possibly erroneous conclusions would be to deduce that the *TriatXIP* gene probably does not perform essential physiological functions in the development life cycle and/or in the responses to environmental changes and/or mycoparasitism of *T. atroviride*. That can be reminiscent of the fact that XIPs represent the smallest fungal MIP subgroup and they seem to be frequently missing in certain fungal clades [20,21,22,23,24,25,26]. However, this would not take into account the fact that mutants have significantly deviant *C*- and *N*-metabolisms, and they fail to produce abundant and classic chlamydospores under ad hoc conditions.

All these results increase the sense of plausible interconnected compensating effects between *Triat*MIP from a same subgroup and counterparts belonging to the three different subgroups, in particular to facilitating common transport across biomembranes of water and a variety of low molecular weight solutes which need to be identified. As a complement, these compensating events do not exclude the possibility of the divergent function of certain MIP members belonging to each subgroup, but whose combination remains essential for the metabolic integrity of the organism: first by the amount of permeable activity from each protomer, but also for the tetramer structures (homo- and hetero-oligomers) where permeability activity can be potentiated when protomers interact with each other in a cooperative mechanism, as observed for yeast, plant and mammal MIP [121,122,123,124]. To the best of our knowledge, there is no homo- or hetero-oligomer complex described in filamentous MIP, opening a vast new field to be further explored.

In terms of solute permeation and foremost with regards to water, the apparent contradiction between the constitutive accumulation of *TriatXIP* transcripts that can be modulated under various biological conditions and the lack of noticeable important growth reduction between ∆*TriatXIP* and *T. atroviride* wild type strains in presence of stressors may be explained by an overlap in water specificity of *Triat*XIP with other MIPs. Because *Triat*XIP and the two co-expressed *Triat*AQPs (three paralogs phylogenetically linked) (Figure 6b) are co-expressed (in particular *Triat*AQP-31598 which displays similar transcriptional patterns to *TriatXIP* under stresses), we can conjecture that these genes are jointly required for cell tolerance and mycoparasitism in *T. atroviride*, and that the function of *Triat*XIP protein may be complemented by AQP-31598 in ∆*TriatXIP* strains. In *S. cerevisiae*, some combined activities of orthodox AQPs and the aquaglyceroporin Fps1 have been suspected to be essential for adjusting appropriate osmotic balance, in particular with regard to hypo-osmotic conditions [125,126]. Interestingly, in addition to playing an integral role in water transport, MIP have also been considered to be potential components in sensing osmotic changes in microorganisms, plants or mammals [127,128,129,130,131]. All fluctuating contexts that modulate MIP expression may result in a modification in plasma membrane water permeability concomitantly to a change in membrane surface tension [132]. Ultimately, because each MIP can be considered a piece maintaining the global cellular homeostasis puzzle, the decrease in ∆*TriatXIP* growth (slightly exacerbated under stresses) may result in complex osmoregulatory imbalances cumulating deviances in water availability and membrane surface tension.

In addition to water, XIPs are involved in transporting small solutes in plants, such as glycerol, urea, ammonia, H_2_O_2_, boron, arsenic, antimony and silicon [30,33,34,83]. In terms of multifunctional permeant-channels, these overlap mechanisms can involve transporting specific solutes which play important roles in nutrient uptake and osmoregulation, such as certain linear polyols (Glycerol, Mannitol, Arabitol, Xylitol, Sorbitol) and nitrogen (Urea, NO_3_^−^, NH_4_^+^), to name just a few [133,134], or certain unexpected permeants considered harmful or toxic at high dosage such as arsenite, antimonite and various byproducts of the glycolysis and tricarboxylic acid cycle (Ethanol, Acetate, Lactate, Malate) [121,135,136,137]. Interestingly, ∆*TriatXIP* exhibits contrasted growth phenotype in the presence of some of these elements such as glycerol, and plausibly with Xylitol (*p* = 0.151) (Figure 5 and Appendix A). Conversely, no impact can be seen with Lactate, Malate, or Erythritol. This might imply the plausible transport of these two polyols, Glycerol and Xylitol. However, these data contrast with the structure modeling simulations that tend to demonstrate that the hourglass of *Triat*XIP presents a pore constriction of 1.86 Å in diameter (which corresponds to the diameter of a water molecule). These physicochemical features are too restrictive to enable molecules other than water to pass through. It is therefore reasonable to assume that *Triat*XIP may act in conjunction with specific channels and transporters dedicated to transport these molecules. The example of glycerol diffusion clearly demonstrates this compensating event. Fps1-like and Yfl054-like aquaglyceroporins are two AQGP subgroups present in yeasts and some filamentous fungi [18,19,20,21,22,23,24,25,26]. Although the Fps1-like function has not yet been fully understood, it is a glycerol-facilitator significantly regulated by osmotic changes. Also, Fps1-like protein has a constriction region above 3.4 Å wide and is defined by more hydrophobic residues, as observed for several GLP and AQGP from *T. harzianum* [28,29,30,31,32,33,34,35,36,37,38,39,40,41,42,43,44,45,46,47,48,49,50,51,52,53,54,55,56,57,58,59,60,61,62,63,64,65,66,67,68,69,70,71,72,73,74,75,76,77,78,79,80,81,82,83,84,85,86,87,88,89,90,91,92,93,94,95,96,97,98,99,100,101,102,103,104,105,106,107,108,109,110,111,112,113,114,115,116,117,118,119,120,121,122,123,124,125,126,127,128,129,130,131,132,133,134,135,136,137,138]. In numerous eukaryotic and prokaryotic organisms, glycerol acts like a carbon source and a major protectant compatible solute under low a_w_ in high osmolarity environments [139]. In response to fluctuating environments where cell osmolarity is impacted, Fps1-like protein can mediate the entrance and release of Glycerol into the cell. However, Fps1-like-deleted cells are able to grow in media with glycerol as the sole carbon source, suggesting a second role of Fps1-like on glycerol uptake [140]. Similarly, GUP1 and GUP2—two active glycerol transporters—play prominent roles in glycerol fluxes in yeasts, complementing Fps1-like protein absence [141]. These compensating occurrences involve a plethora of organic molecules whose transport can be ensured by several transporters and/or channels [142]. In our work, the observed deviant phenotypes of ∆*TriatXIP* transformants may result from indirect consequences of the capture and/or metabolism of these large solutes where the homeostatic regulatory mechanisms inherent to the hydric status had to be significantly altered by the lack of *Triat*XIP. In any case, the loss of XIP activity is not lethal to the organism, possibly due to compensation events, notwithstanding the fact that the chlamydosporogenesis process is deeply impacted.

## 4. Conclusions and Perspectives

Water is the main component of all living cells, and its flow is a fundamental property of life. The MIP superfamily, including the fungal-XIP, is an active candidate in the complex network of multifunctional and/or specific channels and transporters, and this ensemble modulates the high velocity of water exchange across the biological membrane inherent to most organisms’ physiological survival processes. Together with the recently published results [26,29,31,35], our present results complement our previous works on Trichoderma MIP [28] and give a deeper insight into the fungal XIP subgroup and its possible regulation and involvement in the nutritional and mycoparasitic processes of *T. atroviride*, alongside the activities of its paralogue aquaporins and aquaglyceroporins. It is essential to consider that one of the direct consequences of the XIP loss-of-function is its effect on the expression of the other expressed *MIP*-encoding genes of *T. atroviride*, some being up-regulated and others down-regulated. Kinetics in the growth of both the mutant strains and the wild type occurred quite similarly, though delayed for the mutants, and in this respect, the *Triat*MIP transcriptional differences are not the consequence of variable rates of nutrient uptake. We therefore conclude that the fungal *XIP* gene—through the function of its related proteinaceous product—is an important variable in the balance of MIP gene expression in the fungal growth or the cell responses to changing environmental conditions in general, and in particular, during chlamydosporogenesis. However, *Trichoderma* spp. are organisms with specialized lifestyles. An attractive research priority for the future would be the prediction of different XIP ecological functions from different fungal species, supported by extensive structural analyses on the diversity of the sequences involved. Such multidisciplinary analyses will allow a better understanding of the physiological roles of this singular MIP subclass and their functioning in various physiological mechanisms in fungi.

## Figures and Tables

**Figure 1 biomolecules-11-00338-f001:**
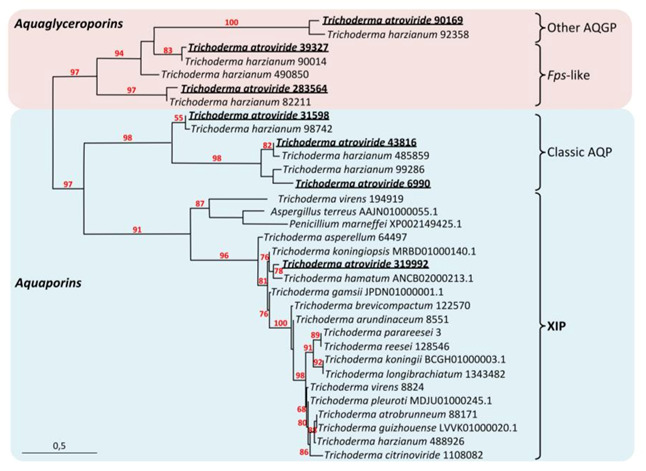
Phylogenetic analysis of *XIP* from the *Trichoderma* genera and selected fungal MIP. AQPs, classical aquaporins; AGPs, aquaglyceroporins; XIP, X-intrinsic protein. Red and blue squares indicate aquaglyceroporin and aquaporin subfamily nodes, respectively. The bootstrap values indicated at the nodes are based on 1000 bootstrap replicates. Branch values lower than 50% are hidden. The distance scale denotes the evolutionary distance expressed in number of nucleic acid substitutions per site. *MIP* sequences from *T. harzianum* (CBS 226.95 v1.0 as reference, JGI) were used as intraspecific out-group references [28], and *XIP* sequences from *Aspergillus terreus* and *Penicillium marneffei* were used as interspecific out-group references [29]. *AQP*, *AQGP* and *XIP* sequences and accession numbers attached after each species name are listed in Appendix A.

**Figure 2 biomolecules-11-00338-f002:**
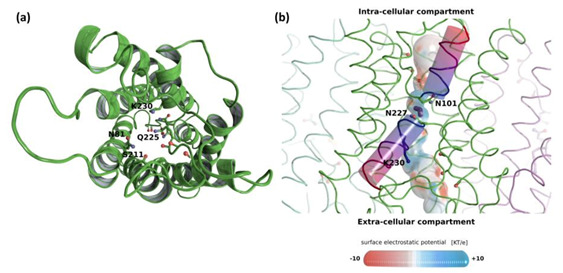
Structural analysis of *Triat*XIP. (**a**) Schematic representation of one protomer of *Triat*XIP as seen from the extra-cellular compartment. The backbone is represented in a diagram and the pore lining residues in sticks. Residues of the ar/R SF constriction are labeled. K230 is unusual as it replaces the very conserved Arginine of the constriction. (**b**) Schematic representation of the tetramer with a focus on one protomer for which the pore surface is figured. Electrostatic potential (KT/e) was computed with APBS (the atoms were previously typed with the PARSE forcefield in PDB2PQR) and projected on the pore surface. The channel surface has a typical positive potential at its NPA region, where the two hemi-helices (represented by cylinders) meet and in the extra-cellular vestibule because of the positively charged lysine 230 side-chain. A water molecule is figured in purple interacting with the two Asparagines of the NPA region. Local electronegative potentials are located at the interaction sites for water molecules with carbonyl groups of the backbone. The two Asparagines of the NPA region, the Lysine 230 of the ar/R constriction and the carbonyl groups of pore-lining residues are represented with sticks. All representations were made with PyMol software [54].

**Figure 3 biomolecules-11-00338-f003:**
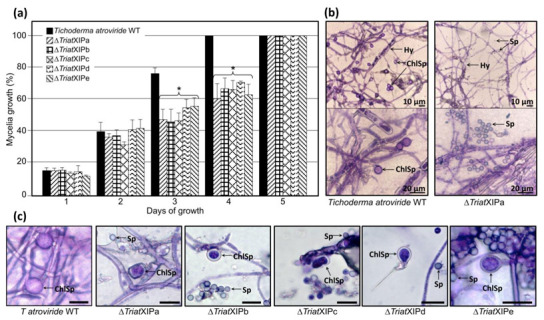
Phenotypes of *T. atroviride* wild type and the five ∆*Triat*XIP mutants upon growth on PDA medium. (**a**) The mycelial growth corresponds to the mean of three independent biological experiments, and bars represent the biological standard deviation. *, Data are statistically significantly different between the wild type and each transformant, as verified by one-way ANOVA analysis followed by Tukey’s post hoc test (*p <* 0.05). (**b**) Light microscopy of hyphae and spores of the parental strain and the ∆*Triat*XIPa mutant for illustration after 20 days of cultivation on PDA media in darkness at 23 °C. The scale bar represents 10 or 20 μm. Samples were colored with toluidine blue (1% *w*/*v*) for 1 min prior to the observations. ChlSp, Chlamydospores; Sp, spores; Hy, hyphes. Each ∆*Triat*XIP are shown in Appendix A. (**c**) Example of morphology of Chlamydospores (ChlSp) and conidiospores (Sp) from the parental strain and the five ∆*Triat*XIP mutants after 14 days of cultivation on corn flour medium (corn flour 62.86 g/L, glycerol 7.54 mL/L, pH 4.2) in darkness at 23 °C. The scale bar represents 10 μm. Samples were colored with toluidine blue (1% *w*/*v*) for 1 min prior to the observations by light microscopy.

**Figure 4 biomolecules-11-00338-f004:**
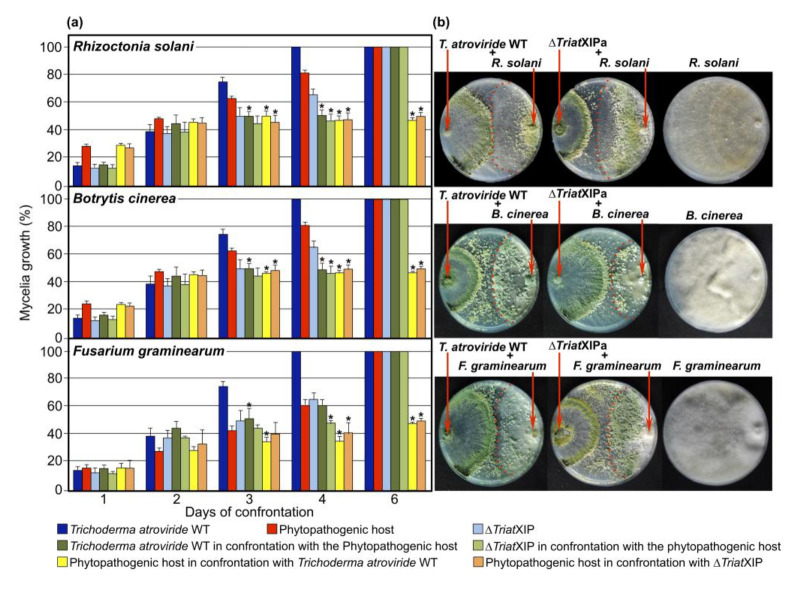
Mycoparasitic activity of the five ∆*Triat*XIP mutants and *T. atroviride* wild strain against *Rhizoctonia solani*, *Botrytis cinerea* and *Fusarium graminearum* as phytopathogenic hosts. (**a**) The mycelial growths on PDA medium correspond to the mean of three independent biological experiments, and bars represent the biological standard deviation. *, Data of the five ∆*Triat*XIP growth rates, and each wild strain (antagonist and phytopathogenic hosts) placed in confrontation situations were statistically and significantly different from that of each strain upon stress-free cultivation at the corresponding time point, as verified by one-way ANOVA analysis followed by Tukey’s post hoc test (*p <* 0.05). Statistical calculations carried out included (*T. atroviride* WT + Host WT vs. *T. atroviride* WT), (∆*Triat*XIPs + Host WT vs. ∆*Triat*XIPs), (Host WT + *T. atroviride* WT vs. Host WT), and (Host WT + ∆*Triat*XIPs vs. Host WT). (**b**) Plate confrontation assays of the *T. atroviride* parental strain (first plate) and the ∆*Triat*XIPa mutant (second plate; ∆*Triat*XIPa is used for illustration, all confrontations implicating the five ∆*Triat*XIPs are presented in Appendix A, and the phytopathogenic hosts (third plate). Pictures were taken six days after inoculation of the two fungi on opposite sides of the plate.

**Figure 5 biomolecules-11-00338-f005:**
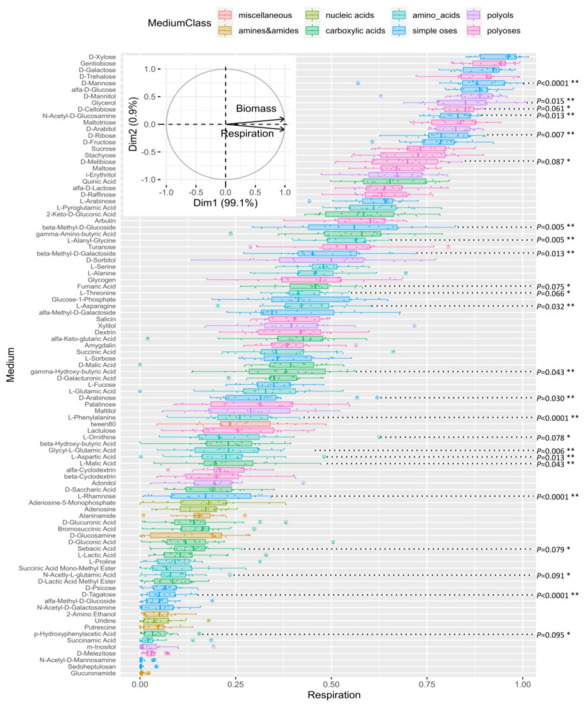
Comparative carbon source utilization profiles of the five ∆*Triat*XIP mutants and the *T. atroviride* parental strain. Strains were grown on 95 carbon sources (FF-plates) using the Biolog Phenotype Microarray system. The order of the organic sources is the rank of respiration rate and substrate use measured at OD490 nm on 95 organic sources and water after 96 h of incubation. Each organic source is colorized according to biochemical class. The mycelial respiration corresponds to the mean of three independent biological experiments per strain, and bars represent the biological standard deviation. ** Respiration rates from the five ∆*Triat*XIP that were significantly different from that of the *T. atroviride* parental strain as verified by *t*-test (*p <* 0.05). * Statistical data not significantly different (*p* > 0.05) between the five ∆*Triat*XIP mutants and the parental strain, but reflect interesting metabolic trends. Statistical data are detailed for the 95 carbon sources in Appendix A.

**Figure 6 biomolecules-11-00338-f006:**
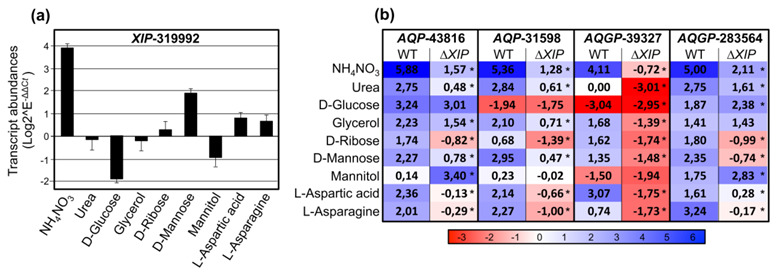
Relative transcription ratios of the expressed MIP genes from *Trichoderma atroviride* wild type and the five ∆*Triat*XIP mutant upon growth on PDA medium supplemented with various inorganic and organic compounds. (**a**) Differential XIP transcript accumulation in the wild strain. (**b**) Differential AQP and AQGP transcript accumulation in the wild type and the five ∆*Triat*XIP mutants. Differential transcript levels for each gene were estimated using real-time qRT-PCR analyses, and normalized by the expression of five housekeeping genes. Relative transcript abundance rates were obtained by the *E*^-ΔΔC*t*^ method. Data correspond to means of three biological replicates for the wild strain and the mean of the five ∆*Triat*XIP mutants. Bars represent the biological standard error. *, Data are statistically significantly different between the wild and the five ∆*Triat*XIP strains, as verified by one-way ANOVA analysis followed by Tukey’s post hoc test (* *p <* 0.05). Expression levels are displayed by the depth of color, where red represents down-regulated regulations and blue denotes up-regulated regulations. Statistical analysis showing the significant difference between wild type and mutants is detailed in Appendix A.

**Table 1 biomolecules-11-00338-t001:** Nomenclature and protein properties of *Triat*MIPs from *Trichoderma atroviride*.

^a^ Loci Proposed Gene Name/Locus	Size (aa)	^b^ MW (kDa)	^b^ pI	^c^ TMH	^d^ SubCL	^e^ NPA		^f^ ar/R SF
						LB	LE	
**Fungal Uncategorized *X*-intrinsic proteins (XIPs)**								
*Triat*XIP-319992	301	32.103	8.75	6	PM	NPT	NPA	N-S-Q-K
**Aquaporin (AQGPs)**								
*Triat*AQGP-31598	276	30.132	7.70	6	PM	NPA	NPV	Y-M-A-R
*Triat*AQGP-43816	312	32.729	6.60	6	PM	NPA	NPV	F-H-T-R
*Triat*AQGP-6990	303	32.453	9.14	6	PM	NPA	NPA	F-H-T-R
**Aquaglyceroporin (AQGPs)**								
*Triat*OtherAQGP-90169	344	37.57	6.75	6 (5) *	PM	SPA	NMA	F-A-T-R
*Triat*FpsAQGP-39327	299	32.653	7.60	6	PM	NPA	NLA	W-G-Y-R
*Triat*FpsAQGP-283564	594	65.625	5.88	6	PM	NPT	NPS	W-T-A-R

^a^ Loci, Protein IDs and AQP types are based on JGI assembly. ^b^ MW, Protein molecular weight; *p*I, protein isoelectric point. ^c^ TMH, Number of transmembrane helices predicted by TMHMM and SOSUI analysis tools; * regions adjusted according to alignment with characterized orthologs [28,29]. ^d^ SubCL, Predicted subcellular localization using Wolfpsort server: PM, plasma membrane; Details in Appendix A. ^e^ NPA, Asparagine, Proline, Alanine. ^f^ ar/R SF, ar/R Selectivity Filter (H2-H5-LE1-LE2).

## Data Availability

All relevant data are within the paper and its Appendix A.

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
