# Peer review of "Fungal X-Intrinsic Protein Aquaporin from Trichoderma atroviride: Structural and Functional Considerations"

_biomolecules, 2021, doi:10.3390/biom11020338_

Round 1
Reviewer 1 Report
In their article “Fungal X-Intrinsic Protein aquaporin from Trichoderma atroviride: structural and functional considerations” the authors investigated the diversity, structure, expression and functional characterization of the fungal XIP from the mycoparasitic species Trichoderma atroviride using different in vitro and in silico methods. Furthermore, they frame a larger picture of why it is interesting to look at the respective protein family and its functional impact. I definitely agree with the authors that aquaporins are of great interest and importance across all forms of live due to their role of keeping the water balance across biological membranes.
Major concern:
I would hesitate showing pf values of 10ns MD simulations of AQP homology models. Even for solved structures several 100ns are state of the art. Considering that not all side chains might be in an optimal orientation I would say that in this case 1000ns would be state of the art. From an 10ns one can solely learn that the homology model is in the open state but not deduce accurate permeability values. Hence, I would either perform longer simulations and monitor the rate over time or do not report any pf value and just show that it is modeled in the open state.
Minor comments:
- Reference supplementary figures in the main text.
- It would be good to have the variety of pore lining residues or at least of the two selectivity filters as a table in the main article.
- Based on the pore lining residues it might be possible to speculate about possible substrates as ammonia, …
- How rare is a lysine in the ar/R constriction of AQPs? Are any other AQPs known with a Lys in the selectivity filter?
Reviewer 2 Report
The present article explains the significance of XIP aquaporin in filamentous fungi T. atroviride. Although plenty of research has been identified XIPs in fungi and plants, this uncategorized intrinsic protein subfamily's functional relevance is still not completely understood.
This article deals with the comprehensive studies related to T. atroviride mutants of XIP. Overall the manuscript is well written and well- presented.
However, some sentences need to be written more clearly.
Some examples are:
-Line 24: "It facilitates transport across biomembranes of water and low molecular weight solutes." can be written like "It facilitates the transport of water and low molecular weight solutes across biomembranes."
-Line 30: "Electrostatic profiles of TriatXIP resemble those of strict aquaporins, predicting water diffusion and possibly other small polar solutes due to particular wider ar/R constriction with a Lysine substitution at the LE2 position." can be written more clearly
Lengthy sentences need commas or shorter sentences for clear understanding.
Some examples are:
-Line 100
-Line 334
Figures:
The format of Figure 1 is not good in pdf file, believing that the publishing department will take care of the issue.
Fonts of Figure 2b is not very clear on the lower side.
Figure 3c shows the morphology of Chlamydospores in the XIP mutants. I would suggest including the morphology of Chlamydospores of the wild- also type for comparison.
Author Response
Please, see the attachment.

Reviewer 3 Report
The manuscript entitle “Fungal X-Intrinsic Protein aquaporin from Trichoderma atroviride: structural and functional considerations” shows the in silico characterization of TriatXIP and phenotype of T. atroviride lacking TriatXIP regar it growth rate, the ability to consume carbon and nitrogen sources and the expression of other T. atroviride MIPs.
This study present a very exhaustive methodology with a lot of work that which sadly could not be exploited for lack of a clear objective.
The introduction is well written but when but when it is read, it is clear that the authors have no concrete objectives since the expose exhaustively the state of the art but there is not a hypothesis and there is not a priori reasons or clues to do the majority of the assays. For example, what the author expected from the assays of Screening for antagonistic activity of T. atroviride? What in the literature could it lead us to believe that the behaviour of the ATraitXIP strains would be different from wild strains? Or for example Why we expected difference in consumption of different sources? Why this sources? What the author expected a priori from this assay?
This is the reason that after a lot of hardly work the author was not able to conclude almost nothing since given that although there is a conclusions section, most of the sentences in this section cannot be considered conclusions (see comments below).
In general Result and Discussion section it is too long and the author try to explain something that not even they themselves understand, and so they go round in circles without getting anywhere.
I suggest that it is worthy that after this hardly work the authors should rethink their objectives and hypotheses to make the article clearer so that its full potential can be exploited.
Some specific coments:
Line 55: whether or where?
Line 106-108: So far, XIP has been found to be the only MIP subgroup common to both plant and fungal lineages, forming a plausible monophyletic group distinct from the other MIP subfamilies. This suggests specific roles in solute transport and/or protein regulation of these channels which need to be better understood.
This sentence is not referenced. Then is a kind of author conclusion? How have they inferred that fungal and plant XIP is a monophyletic group? In addition, the author should take into account that fungi and plant are unrelated phylogenetic groups.
Line 128-129: Taken together, our results indicated that TriatXIP acts along with other MIP couterparts to regulate vegetative morphogenesis, stress responses, and sporogenesis in T. atroviride.
That sentence is a kind of conclusion and should therefore go at the end of the discussion.
In Figure 1 some question sings are in the middle of the names
Headline 3.3. and 3.4. should be merged.
Headline 3.6 and 3.7 should be merged
The authors should consider that certain aquaporins are able to facilitate the passage of oxygen with certain change in respiration Zwiazek, JJ; Xu, H; Tan, X; Navarro-Ródenas, A; Morte, A. 2017. Significance of oxygen transport through aquaporins Scientific Reports. 7, pp.40411.
717-722 to find difficult to correlate N and C sources with XIP expression strain is due to no initial hyphothesis was proposed and this assay was made with no clear objective.
873-876. If that is an hypothesis should be write at the end of the introduction. Nevertheless, this hypothesis is very unspecific with no specific question that could be answer.
886-888: we conclude that the fungal XIP gene - through the function of its related proteinaceous product - is an important variable in the balance of MIP gene expression in the fungal growth or the cell reponses to changing environnemental conditions.
That sentence is not a conclusion is just and observation since there is not interpretation of the results. This sentence just describes the results.
889-895 it is not a conclusion this is something that we know before this paper
899-901 it is not a conclusion derived from the result from this paper it this sounds like a publicity thing.
Author Response
Please, see the attachment.

Round 2
Reviewer 3 Report
The manuscript has improved markedly and has gained some sense to the reader, although it is a pity that much of the hard work done, it has not been fully exploited.